# Recent Development of Aminoacyl-tRNA Synthetase Inhibitors for Human Diseases: A Future Perspective

**DOI:** 10.3390/biom10121625

**Published:** 2020-12-01

**Authors:** Soong-Hyun Kim, Seri Bae, Minsoo Song

**Affiliations:** New Drug Development Center (NDDC), Daegu-Gyeongbuk Medical Innovation Foundation (DGMIF), 80 Cheombok-ro Dong-gu, Daegu 41061, Korea; soong@dgmif.re.kr (S.-H.K.); bsr4202@dgmif.re.kr (S.B.)

**Keywords:** aminoacyl-tRNA synthetase, small molecule inhibitors, human diseases

## Abstract

Aminoacyl-tRNA synthetases (ARSs) are essential enzymes that ligate amino acids to tRNAs and translate the genetic code during protein synthesis. Their function in pathogen-derived infectious diseases has been well established, which has led to the development of small molecule therapeutics. The applicability of ARS inhibitors for other human diseases, such as fibrosis, has recently been explored in the clinical setting. There are active studies to find small molecule therapeutics for cancers. Studies on central nervous system (CNS) disorders are burgeoning as well. In this regard, we present a concise analysis of the recent development of ARS inhibitors based on small molecules from the discovery research stage to clinical studies as well as a recent patent analysis from the medicinal chemistry point of view.

## 1. Introduction

Aminoacyl-tRNA synthetases (ARSs) are a family of 20 essential enzymes that ligate amino acids to their corresponding tRNAs and translate the genetic code during protein synthesis [1]. The two-step catalytic reaction of ARSs involves the formation of an enzyme-bound aminoacyl-adenylate (AMP-aa) followed by the formation of an aminoacylated tRNA (tRNA-aa) by transferring the amino acid to the corresponding tRNA. This catalytic reaction of ARSs plays a pivotal role in protein synthesis, which is essential for the growth and survival of all cells (Figure 1). ARSs have long been studied as therapeutic targets for pathogen-derived infectious diseases, predominantly because of concerns that ARS catalytic site inhibitors would block translation in normal cells [2,3,4]. In this regard, various seminal reviews have discussed ARS inhibitors for the development of antibiotics from different perspectives [5,6,7,8,9].

However, a significant amount of research targeting mammalian ARSs for the treatment of human diseases have recently emerged. For example, febrifugine derivatives, including halofuginone which targets prolyl-tRNA synthetase (ProRS), are under investigation for the treatment of cancer, fibrosis, and inflammatory diseases [10,11]. Additionally, threonyl-tRNA synthetase (ThrRS) was identified as a target protein of antiangiogenic borrelidin [12], and a peptide fragment derived from tryptophanyl-tRNA synthetase (TrpRS) was studied as a possible anti-angiogenic agent [13]. However, Pfizer has dropped its development of the TrpRS fragment as an anti-angiogenic lead a while ago. Accordingly, Kwon et al. [14] separately provided thorough reviews on the physiology, pathology, and structural aspects of ARSs, as well as the development of ARS inhibitors for human diseases, including infectious diseases, cancer, inflammatory diseases, and central nervous system (CNS) disorders. In this review, we focus on the recent development of ARS inhibitors ranging from discovery to clinical trials from medicinal chemistry perspectives and provide a related patent analysis.

## 2. Structural Class of ARS Inhibitors

### 2.1. Benzoxaboroles

Benzoxaboroles are a class of boron-containing compounds with a broad range of biological activities. A subset of benzoxaboroles have antimicrobial activity primarily due to their ability to inhibit leucyl-tRNA synthetase (LeuRS) via the oxaborole tRNA-trapping (OBORT) mechanism, which involves the formation of a stable tRNA^Leu^−benzoxaborole adduct in which the boron atom interacts with the 2′- and 3′-oxygen atoms of the terminal 3′ tRNA adenosi006Ee [15]. In a related study, researchers at Anacor Pharmaceuticals reported an X-ray crystal structure of *Thermus thermophilus* LeuRS complexed with tRNA^Leu^ and the benzoxaborole-based LeuRS inhibitor AN2690 to form a tRNA–AN2690 adduct possessing tetrahedral spiroborate structure **1** (Figure 2) [16]. The tetrahedral spiroborate adduct was stabilized by the formation of covalent bonds between the Lewis acidic boron atom and the 2′- and 3′-oxygen atoms of the tRNA’s 3′-terminal adenosine as a nucleophile [17,18]. The oxaborole scaffold can reversibly form covalent tetrahedral complexes with nucleophiles, such as hydroxyl groups, because of the presence of the heterocyclic boron atom, which has an empty *p* orbital. Targets for these compounds include β-lactamases, cyclic nucleotide phosphodiesterase 4 (PDE4), Rho-associated protein kinase (ROCK), carbonic anhydrase, and LeuRS [19].

A recent example of the benzoxaborole class of inhibitors is the nitrophenyl sulfonamide-based benzoxaborole PT638; Si et al. investigated PT638 to determine its mode of action [15]. The finding that PT638 has potent antibacterial activity against methicillin-resistant *Staphylococcus aureus* (MRSA) (MIC [minimum inhibitory concentration] of 0.4 μg/mL) and does not inhibit *Staphylococcus aureus* LeuRS (*Sa* LeuRS) (IC_50_ [the half maximal inhibitory concentration] >100 μM) suggests the possibility that it has targets other than LeuRS. The selection for resistance to PT638 using MRSA and whole-genome sequencing of three colonies that showed MICs of ≥3.125 μg/mL resulted in three mutations in the LeuRS gene, including D343Y, G302S, and F233I, all of which are located within the editing domain of *Sa* LeuRS, thereby confirming *Sa* LeuRS as the target of PT638. A metabolism study of PT638 using MRSA cell lysate and a SAR (structure-activity relationship) study of PT638 analogs revealed that PT638 might be a prodrug that can be reduced to its amine analog PT662 mainly by nitroreductase NfsB in MRSA (Figure 3).

In a related study, GlaxoSmithKline (GSK) developed a series of 3-aminomethylbenzoxaborole derivatives **4** and **5** that target *Mycobacterium tuberculosis (Mtb)* LeuRS [20]. The boron atom in the molecules was critical for *Mtb* LeuRS activity since it forms a bidentate covalent adduct with the terminal nucleotide of tRNA, Ade76, and traps the 3′ end of tRNA^Leu^ in the editing site to inhibit leucylation and thus protein synthesis. The (*S*)-aminomethyl side chain at C-3 in **4** and **5** was critical for binding, and C-4 halogen atoms (Cl and Br) significantly improved *Mtb* LeuRS activity, antitubercular activity against *Mtb* H37Rv, and selectivity against other bacteria. However, potential toxicity issues from the inhibition of mammalian cytoplasmic LeuRS and tolerance issues of **5** in a once daily dose of 50 mg/kg in a mouse acute tuberculosis (TB) infection model caused serious concerns for its use as a TB drug in long-term therapy. Throughout a series of medicinal chemistry campaigns to examine the structure–activity relationship (SAR) of 3-aminomethylbenzoxaborole derivatives, compounds **6**–**10** were found to improve both selectivity against human cytoplasmic LeuRS and HepG2 cell toxicity compared with **4** and **5**, presumably by decreased lipophilicity and increased polarity in the molecule (Figure 4) [21]. Triaging lead compounds through in vivo pharmacokinetic (PK) analysis and efficacy assays using acute and chronic mouse TB infection models identified **8** (GSK656) as a preclinical candidate for tuberculosis; **8** successfully completed its first-time-in-human (FTIH) study to evaluate the safety, tolerability, pharmacokinetics, and food effects of single (5, 15, and 25 mg) and repeat oral doses (5 and 15 mg, 14 days, qd [quaque die, once a day]; NCT03075410) [22]. Moreover, **8** is currently in a phase 2 clinical trial for drug-sensitive pulmonary tuberculosis to evaluate its early bacterial activity, safety, and tolerability (NCT03557281). Details of the clinical studies of **8** will be discussed in a later section.

### 2.2. Cladosporin

Cladosporin isolated from *Cladosporium cladosporioides* and *Aspergillus flavus* was found to have potent antimalarial activity against both liver- and blood-stage *Plasmodium falciparum* by targeting the parasite cytosolic lysyl-tRNA synthetase (*Pf* LysRS) and terminating protein biosynthesis [23]. Despite its high clearance and poor oral bioavailability in vivo [24,25], the favorable selectivity and potency shown by cladosporin against *Pf* LysRS over human LysRS spurred medicinal chemistry efforts to further develop it to overcome its metabolic limitations as an oral drug. In this regard, Rusch et al. analyzed cladosporin by incubating it with mouse liver microsomes in vitro and revealed that three major metabolically weak points might cause the metabolic liabilities of cladosporin (Figure 5) [26]. The hydroxyl group, tetrahydropyran, and C-4 of isochromenone were potentially the three major metabolically weak points. Additionally, the X-ray crystal structure of cladosporin and *P. falciparum* LysRS (*Pf* LysRS) suggested an H-bond with Glu332 and the π/cation ‘cage’ from adjacent aromatic and charged residues, and the interaction between the lactone ring and Arg559 through water as the major interactions [27]. Based on these basic findings, medicinal chemistry efforts were focused on the three metabolically weak points of cladosporin to generate the three major derivatives **A** (**12**), **B** (**13**), and **C** (**14**) as representative examples. Upon screening these libraries using the Transcreener AMP assay system to assess aminoacylation by LysRS [23], the unsaturation of the isocoumarin core and transformation of the metabolically unstable tetrahydropyran ring to the cyclohexyl ring yielded a selective LysRS inhibitor with improved metabolic stability at the level of hepatic extraction. In line with a previous report by Rusch et al. [26] focusing on metabolic weak points of cladosporin, Zhou et al. confirmed that the methyl tetrahydropyran moiety of cladosporin could be replaced by a more stable methylcyclohexane, while maintaining potency in the ATP hydrolysis assay (compounds **15** and **16**) [28]. The methyl group in the cyclohexane ring was important for the hydrophobic interaction with Ser344, resulting in increased potency of 4-fold compared to that of compound **15**. The replacement with a lactam group or a conjugated Δ^3,4^ double bond within the isocoumarin ring was tolerated, but the two phenolic hydroxyl groups were critical for binding to LysRS (Figure 5).

Another medicinal chemistry campaign to develop a new antimalaria and anti-cryptosporidiosis drug targeting *Pf* LysRS1 and *Cryptosporidium parvum (Cp)* LysRS was continued by Baragaña et al. using cladosporin as a starting point [29]. Its low druggability due to high metabolic instability was optimized by replacing the isocoumarin core with the chromone scaffold connected to a metabolically stabilized cyclohexane ring with an amide linker, as shown in Figure 6. Compound **21** was active against both *Pf* LysRS1 (IC_50_ = 0.015 μM, luciferase ATP consumption assay) and whole-cell bloodstream *P. falciparum* 3D7 (EC_50_ [the half maximal effective concentration] = 0.27 μM) and was selective compared with *human (Hs)* LysRS (IC_50_ = 1.8 μM). The pharmacokinetic properties were optimized by stabilizing the metabolically labile cyclohexyl ring and blocking the point of potential hydroxylation of the phenyl group in the chromone core. As a result, compound **21** showed good systemic exposure with excellent oral bioavailability (*F* = 100%) and a moderate half-life (*t*_1/2_ = 2.5 h). In vivo efficacy was evaluated using a NODscidIL2Rγnull mouse (SCID) model and doses up to 40 mg/kg (po [per os, oral administration]), and it was found that a daily oral dose of ED_90_ = 1.5 mg/kg (1.0–2.3 mg/kg) (estimated daily exposure in blood AUC_ED90_ [area under the curve at ED_90_] = 11,000 ng·h/mL/d (6900–14,000 ng·h/mL/d)) reduced parasitemia by 90%. A high level of sequence identity within the active site region of *Pf* LysRS1 and *Cp* LysRS (96% identity) and an overall sequence identity of 47.7% and similarity of 64.6% across the entire protein was observed from the evaluation of compound **21** against *Cryptosporidium parvum* both in vitro and in vivo. Compounds for cryptosporidiosis treatment must also have good exposure in the gastrointestinal tract and possibly also some systemic exposure since the parasite is found predominantly in the gastrointestinal tract with some in the biliary tract. In this regard, compound **21** was worthy of consideration for cryptosporidiosis since 17% of the oral dose was found in mouse stools, suggesting that some biliary excretion had occurred. Compound **21** showed efficacy in vivo in two different *Cryptosporidium* mouse models, NOD SCID gamma and INF-γ-knockout, at 20 mg/kg (po). However, the toxicity concern of **21** at 50 mg/kg in mice (po) at which the blood concentration in mice reached the EC_50_ for HepG2 cells limited further development. It is presumed that the toxicity is due to the inhibition of mammalian LysRS.

Considering the significant structural differences between cladosporin derivatives (**11**–**19**) and the newly found chromone derivatives (**20** and **21**), Baragaña et al. analyzed the binding modes of cladosporin and **20** by studying their X-ray co-crystal structure with *Pf* LysRS1 (Figure 7) [29]. Cladosporin binds to *Pf* LysRS1 within the ATP-binding pocket with the isocoumarin moiety occupying the same space as the adenine ring of ATP, and the tetrahydropyran system taking the same position as the ribose ring of ATP. The two phenolic hydroxy groups of cladosporin form hydrogen bonds with E332 and the NH of N339, while the carbonyl interacts with a highly coordinated conserved water molecule. Although **20** binds with *Pf* LysRS1 in the ATP-binding pocket in a similar manner to cladosporin, the bicyclic core is rotated 30° anticlockwise with respect to cladosporin. The chromone core stacks between F342, H338, and R559. The ring carbonyl forms an H-bond to the backbone NH of N339, mimicking the N1 of adenine and the O1 OH of cladosporin. The amide carbonyl H bonds to a highly conserved water molecule coordinated by the side chain of D558 and R559. The cyclohexyl moiety projects into a pocket formed by the side chains of R330, F342, and S344 and the backbone of L555 and G556. This pocket is completed by the substrate lysine and is similar to that occupied by the tetrahydropyran ring of cladosporin, except that the cyclohexyl ring probes deeper into the pocket [29].

### 2.3. Borrelidin

Borrelidin is known as a potent inhibitor of threonyl-tRNA synthetase (ThrRS) for malaria. However, its lack of selectivity against human ThrRS and high toxicity to human cells limit further development as a drug candidate. In this regard, Novoa et al. derived a series of borrelidin analogs that circumvent these potential problems by showing higher selectivity and lower cytotoxicity from minor structural changes in the carboxylic acid side chain [30]. In particular, compound BC196 (**23**), which released rotational constraints by adopting linear carboxylic acid, and compound BC220 (**24**), which converted terminal carboxylic acid to pyridyl ethyl ester on the cylcopentane ring, showed improved selectivity against *P. falciparum* over human HEK293 cells by 13,579-and 4048-fold, respectively. Considering that borrelidin exhibited 355-fold activity difference between *P. falciparum* and human HEK293 cells, selectivity was improved by 11–38-fold. From in vivo efficacy studies using a *Plasmodium yoelii* 17XL-infected mouse model, **23** and **24** showed 100% survival over 20 days after drug treatment at 6 mg/kg per day. In particular, **24** showed complete clearance of parasites in the blood at 6 mg/kg per day, 80% survival, and ~3% parasitemia at 0.25 mg/kg per day. It is also interesting that **24** was not cleaved to borrelidin by esterase, as determined in time course experiments with infected red blood cells (RBCs) over 27 h, suggesting that the biological effect of **24** is not due to its conversion to borrelidin by RBCs or plasmodial esterases (Figure 8).

### 2.4. Benzothiazoles

Unlike other LysRS inhibitors that target the active site of pathogen-derived LysRS, Kim et al. reported a new class of small molecule inhibitors, including YH16899, that bind to human LysRS without affecting the catalytic activity of LysRS [31]. Physiologically, it is known that LysRS binds to the 67-kDa laminin receptor (67LR), a target for antimetastatic therapeutics [32], in the plasma membrane and enhances cell migration that causes cancer metastasis. Thus, it is thought that directly inhibiting the interaction between LysRS and 67LR by a small molecule could suppress cancer metastasis and derive a new class of antimetastatic cancer therapy. In this regard, **25** showed specific inhibition of the binding between the LysRS and 67LR pair based on in vitro pull-down and immunoprecipitation (IP) assays and decreased the amount of membranous 67LR using LysRS-overexpressing H226 squamous lung carcinoma cells (Figure 9). The in vivo efficacy of **25** was demonstrated using three different in vivo mouse models, including (i) a mouse breast cancer model (4T1 cells, 100 mpk [mg per kilogram] and 300 mpk, po, ~60% inhibition of tumor metastasis), (ii) a Tg (MMTV-PyVT) model (100 mpk, po, ~70% reduction of pulmonary nodule formation), and (iii) a cancer cell colonization model (A549 cells expressing red fluorescence protein (RFP), ~50% reduction of tumor metastasis in the brain and bone tumors over 7 weeks). Overall, Kim et al. showed that the protein–protein interaction of LysRS-67LR is a promising approach for the development of new therapeutics for human diseases.

### 2.5. Halofuginone

Halofuginone (HF, **26**), a well-known antifibrotic agent in preclinical and clinical studies, inhibits prolyl-tRNA synthetase (ProRS) activity [10]. Mechanistically, **26** competitively binds to the proline-binding pocket of the catalytic site of ProRS and causes drug resistance due to the accumulation of proline [33]. It is anticipated that the antifibrotic effect of **26** would be diminished in fibrotic tissue because a higher concentration of proline is often observed in fibrotic tissue than in nonfibrotic tissue [34]. In this regard, Shibata et al. from Takeda Pharmaceuticals developed T-3833261 (**27**), a potent ATP competitive inhibitor of ProRS that does not bind to the proline-binding site [11]. The antifibrotic activity of **27** was evaluated both in vitro and in vivo. From the transforming growth factor beta (TGF-β)-induced fibrotic assay, **27** inhibited the expression of fibrotic markers, including α-smooth muscle actin (α-SMA) and type I collagen (Col1a), by reducing mothers against decapentaplegic homolog 3 (Smad3) expression in human skin fibroblasts. These results were directly correlated with the in vivo TGF-β-induced mouse model. The topical application of **27** reduced the expression of fibrotic genes such as Col1a1, Col1a2, and α-SMA (Figure 10).

In a related study, it was presumed that the proline-rich nature of profibrotic proteins such as collagens may be attributed to glutamyl-prolyl-tRNA synthetase (GluProRS)-mediated translational regulation during cardiac fibrosis. Several studies with indirect evidence support the regulatory role of GluProRS as a target gene for cardiac fibrosis. For example, hypoactive mutations in the ProRS domain of GluProRS lead to hypomyelinating leukodystrophy without causing any known cardiac dysfunction in patients, implying that reduced ProRS enzymatic activity may not adversely affect normal heart function while reducing proline-rich protein synthesis [35]. In this regard, Akashi Therapeutics used **26** to treat Duchenne muscular dystrophy (DMD), analogous to reducing fibrosis by **26** and increasing muscle strength [36,37,38,39]. Recently, Wu et al. examined GluProRS-mediated regulatory mechanisms of profibrotic protein synthesis at the translational level and identified increased GluProRS in failing human and mouse hearts [40]. Increased GluProRS contributed to the elevated translation of proline (Pro)-rich (PRR) mRNAs, and genetic knockout of the glutamyl-prolyl-tRNA synthetase gene (*Eprs*) reduced fibrosis in different heart failure mouse models. Additionally, the selective GluProRS inhibitor **26** reduced the translation of PRR mRNAs, such as collagens and other novel PRR genes, including latent TGF-β-binding protein 2 (LTBP2) and sulfatase 1 (SULF1).

### 2.6. Sulfonamido Propanamides and 2-Aminopyrimidines

Aminoacyl-tRNA synthetase-interacting multifunctional protein 2 (AIMP2), a scaffold in the multi-tRNA synthetase complex (MSC), dissociates from the MSC and influences the activity of the p53, TGF-β, tumor necrosis factor-α (TNF-α), and Wingless/int-1 (WNT) signaling pathways [41,42]. A splicing variant of AIMP2 lacking exon 2 (AIMP2-DX2) correlates positively with cancer by compromising the tumor-suppressive function of native AIMP2 through competitive interactions with p53 fuse-binding protein (FBP) and TNF receptor-associated factor 2 (TRAF2) [43]. Lim et al. showed that AIMP2-DX2 binds with heat shock protein 70 (HSP70) and is stabilized by blocking the seven in absentia homolog 1 (Siah1)-dependent ubiquitination of AIMP2-DX2, which leads to cancer progression in vivo [44]. Representative AIMP2-DX2 inhibitors (**28**~**30**) are described in Figure 11. Based on this mode of action of AIMP2-DX2, the protein–protein interaction inhibitor BC-DXI-495 (**28**) was found to specifically inhibit the AIMP2-DX2 + HSP70 interaction and suppress cancer cell growth in vitro (IC_50_ = 4.2 μM, luciferase assay; EC_50_ = 14.2 μM, cell viability assay, A549 cells) and AIMP2-DX2-induced tumor growth in vivo (~50% decrease, 50 mpk, ip [intraperitoneal injection], 2 weeks, H460 stably expressing AIMP2-DX2 WT [wild type], BALB/cSLC mice). Consecutively, Sivaraman et al. reported BC-DXI-843 (**29**), which was optimized from BC-DXI-495 [45]; **29** significantly improved in vitro activity in luciferase and cell viability assays (IC_50_ = 0.92 μM, luciferase assay; EC_50_ = 1.2 μM, cell viability assay, A549 cells). The in vivo efficacy of **29** in the H460 xenograft mouse model was comparable with that of **28** by inhibiting ~60% tumor growth (50 mpk, bid [bis in die, twice a day], ip, 2 weeks). In the same context, Lee et al. separately developed the 2-aminophenylpyrimidine derivative **30** for AIMP2-DX2 interaction inhibition with >80% tumor growth inhibition in an H460 xenograft mouse model [46].

### 2.7. Bicylic Azetidines

Lastly, Kato et al. reported bicyclic azetidines such as **31** and **32** (Figure 12) as potent inhibitors of *Pf* phenylalanyl-tRNA synthetase (PheRS) (EC_50_ = 9 and 5 nM, respectively, against a multi-drug resistant strain, *Pf* Dd2) for the treatment of malaria [47]. Lead optimization campaign allowed the improvement of in vitro potencies of **31** against *Plasmodium cynomolgi* (EC_50_ = 3.34/2.86 μM for small form [SF, hypnozoite-like]/large form [LF, schizont], respectively), *P. falciparum* liver-stage (EC_50_ = 1.31 μM), and *P. falciparum* transmission-stage, gametocyte IV–V (EC_50_ = 663 nM) to EC_50_ = 0.933/1.04 μM (for SF/LF), 0.34 μM, and 0.16 μM in **32**, respectively. Issues in **31** such as poor solubility (<1 μM in PBS), high clearance in human and mouse liver microsomes (Cl_int_ = 142 and 248 μL/min/mg, respectively), and a high volume of distribution in vivo (12 L/kg) were optimized by replacing the hydroxymethyl of **31** to *N*,*N*-dimethylaminomethyl group in **32** to give improved solubility (15 μM in PBS), decreased clearance in human and mouse liver microsomes (Cl_int_ = 31 and 21 μL/min/mg, respectively), and a better volume of distribution in vivo (24 L/kg), resulting in a longer half-life (32 h) which can be suitable for single-dose oral treatment. In vivo efficacy studies of **32** in multiple stages of malaria models including *Plasmodium berghei* CD-1 and *P. falciparum* huRBC NSG mouse models showed complete elimination of blood- and liver-stage parasites by a single dose during the study period. This study has served as an example of developing a cure for malaria as well as the potential to give prophylaxis and prevent disease transmission. Additionally, Vinayak et al. reported that bicyclic azetidine **32** is a potent inhibitor of *Cryptosporidium parvum* PheRS (*Cp* PheRS) for diarrheal disease in young children [48]. The in vivo efficacy of **32** for cryptosporidiosis was shown in immunosuppressed mouse models. Overall, it is thought that bicyclic azetidines such as **32** possess a unique core skeleton and warrant further examination to exploit their chemical space for applications not only in infectious diseases, but also for other diseases.

## 3. Aminoacyl-tRNA Synthetase (ARS) Therapeutics in Patents

As mentioned before, much research on eukaryotic aminoacyl-tRNA synthetases (*e*ARSs) has recently been conducted to determine the different roles of human ARSs other than being the “housekeeping” enzyme for protein synthesis in various human diseases [14]. Human ARSs have also become major targets in the treatment of proliferative diseases, such as cancers, inflammatory diseases, autoimmune diseases, and fibrosis. In this section, we discuss patents related to therapeutic development targeting both prokaryotic and eukaryotic ARSs. Since Gadakh et al. at the Rega Institute for Medical Research reported a patent review on ARS inhibitors as antimicrobial agents in 2012 [7], we specifically analyzed patents from 2013 to the present day. From the SciFinder search, eight relevant patents were identified, and we present them in the alphabetical order of ARS targets.

### 3.1. Glycyl-tRNA Synthetase (GlyRS) Inhibitors

Professors Schimmel and Yang at the Scripps Research Institute disclosed four different glycylsulfamoyladenosine derivatives (GlySAs, Figure 13) as human glycyl-tRNA synthetase (GlyRS) inhibitors for cancer therapy by lowering neddylation [49]. The covalent attachment of NEDD8 (neuronal precursor cell-expressed developmentally downregulated gene 8) to the core cullin protein of cullin-RING ligase E3s (CRLs, e.g., Ubc12), called neddylation, is a crucial process for the ubiquitination of multiple protein substrates mediated by GlyRS. GlyRS is highly expressed in most malignant cancers, such as breast cancer, ovarian cancer, and lung cancer. Selective binding of GlySAs to the catalytic domain of human GlyRS inhibited the attachment of NEDD8, consequently reducing the level of neddylation by more than 50% in various cancer cells. The patent provided the systematic mode of action (MOA) evaluation of GlyRS in the neddylation pathway and the confirmation of GlyRS inhibition with the crystal structure of GlySA (**33**) binding to human GlyRS (PDB code: 2ZT8). GlySAs were screened with the NCI 60 cell panel assay at a single dose of 5–10 M, and GlySA (**33**) showed a range of growth inhibition values (GI_50_ = 0.372–3.715 μM) in 58 cancer cell lines. With the lung metastasis model induced by MDA-MB-231 cells, **33** was administered via intravenous injection twice a week at a dose of 4 mg/kg (*n* = 10). After 10 days of injection of GlyRS inhibitors, **33** reduced lung metastases in the mouse model with 69% efficacy, which was greater than that (4.4 mg/kg, 40% efficacy) of MLN4924. MLN4924, an inhibitor of NEDD8-activating enzyme (NAE), is under development for various types of tumors in a clinical setting [50].

### 3.2. Leucyl-tRNA Synthetase (LeuRS) Inhibitors

Oxford Drug Design Ltd. reported a leucyl-tRNA synthetase (LeuRS) inhibitor from 2-amino-*N*-(arylsulfinyl) acetamide compounds as an antibiotic agent [51]. They prepared a series of 46 compounds and their corresponding TFA (trifluoroacetic acid) salts, as shown in Figure 14, of which the dissociation constants (*K_d_*) were measured for *Escherichia coli* LeuRS by isothermal titration calorimetry (ITC) and were found to range from 1.39 nM to 9130 nM. The antibacterial activity of the 26 compounds was evaluated in seven bacterial strains (ATCC25922, EFFLUX, B1966, ATCC700603, ATCC27853, B1931, and ATCC49247) with MIC values from 0.25 to >64 mg/L, and compound **37** was the most potent compound with MIC values in all strains (0.5, 0.25, 1, 1, 32, 2, and 4 mg/L, respectively) comparable to those of controls, such as ciprofloxacin, ceftazidime, meropenem, and azithromycin. No compound showed any potential cytotoxic effects against the human hepatic cell line HepG2 (ATCC HB-8065).

Jirgensons et al. at the Latvian Institute of Organic Synthesis developed *N*-aminoacyl-arylsulfonamide derivatives as aminoacyl-tRNA synthetase (ARS) inhibitors for the treatment of bacterial infections [52,53]. Arylsulfonyl chlorides were converted to the corresponding arylsulfonamides, in which the terminal amine was coupled with amino acids to provide 52 compounds of *N*-aminoacyl-arylsulfonamides, as shown in Scheme 1. All compounds were tested for in vitro enzyme inhibition assays against leucyl-, valyl- and isoleucyl-tRNA synthetases (LeuRS, ValRS, and IleRS, respectively) from *E coli* and *S. aureus* and showed IC_50_ values from 0.0034 to >50 μM for *E.*
*coli* LeuRS and from 0.26 to >50 μM for *S.*
*aureus* LeuRS. However, none of the compounds presented any inhibitory activities at concentrations of 50 μM or below against IleRS and ValRS bacterial enzymes from either *E.*
*coli* or *S.*
*aureus*. Therefore, the scope of substituents on the phenyl ring was investigated further. It was found that the compounds bearing a meta-substituted phenyl ring as well as a leucine moiety presented a significant increase in the inhibition of LeuRS from *E.*
*coli* and *S.*
*aureus* with submicromolar activity (Figure 15) (e.g., IC_50_ values of compound **42**: 0.002 μM for *E.*
*coli* LeuRS and 0.33 μM for *S.*
*aureus* LeuRS).

### 3.3. Methionyl-tRNA Synthetase (MetRS) Inhibitors

The Buckner group at the University of Washington found that compounds bearing *N*-containing heteroaromatic bicyclic rings inhibit methionyl-tRNA synthetase (MetRS) for antibacterial treatment [54,55]. A total of 249 compounds with four major different scaffolds were synthesized and tested in 13 different in vitro assays against MetRS (*Trypanosoma brucei*) ATP depletion (A), MetRS (*S. aureus*) ATP depletion (B), MetRS (*Brudella*) ATP depletion (C), MetRS (*Giardia*) aminoacylation (D), *T. brucei* (E), *Trypanosoma cruzi* (F)*, Giardia intestinalis* (ATCC50508) (G), *Leishmania*
*amazonensis* (H)*, P. falciparum* (3D7) (I), *S. aureus* (ATCC29213) (J), *S. aureus* MRSA (ATCC43300) (K), *Enterococcus faecalis* (ATCC29212) (L), and *Enterococcus faecium* (ATCC51559) (M). Two compounds bearing the same scaffold of pyridoimidazole, **45** and **46**, showed excellent potency in bacterial MetRS inhibition (IC_50_), bacterial growth inhibition (EC_50_), and MIC values in most of the in vitro cell assays (Figure 16).

The Martin group at the University of Texas at Austin reported *T. brucei* methionyl-tRNA synthetase (MetRS) inhibitors for the treatment of African trypanosomiasis and sleeping sickness [56,57]. Two major classes of compounds incorporating tetracyclic tetrahydro-beta-carboline and 4-amino-2-piperidone were prepared to selectively inhibit the MetRS of *T. brucei*. The inhibitory activities of three tetracyclic carboline compounds (**47**–**49**) were tested for the *T. brucei* MetRS aminoacylation assay, showing submicromolar IC_50_ values as depicted in Figure 17. One of the tetracyclic compounds exhibited no off-target activity when screened against a panel of 45 CNS proteins.

Additionally, the internal C-C bond of piperidine was removed to yield 37 4-amino-2-piperidones, including **50**–**61,** which were screened for *T. brucei* growth inhibition at a single dose (1 μM or 10 μM). Eleven compounds (**50**–**60**) showed decreased activity in the *T. brucei* growth inhibition assay by showing more than 80% inhibition at 10 μM; **61** with a methoxy on the C-5 of the indole ring showed better activity with ~60% inhibition at 1 μM (Table 1).

### 3.4. Phenylalanyl-tRNA Synthetase (PheRS) Inhibitors

The Kahne group at Harvard University developed tRNA synthetase inhibitors of secondary amines for killing Gram-negative bacteria and treating tuberculosis [58]. Although the target enzyme is not specifically mentioned in their patent, phenylalanyl-tRNA synthetase (PheRS; FARS, formerly FRS) might have been explored due to the specific note about FRS in the table of antimicrobial activity. They synthesized 458 secondary amines, one tertiary cyclic amine, 15 tertiary cyclic amides, and one tertiary cyclic carbamate compound. The antimicrobial activities of each compound were measured via the Gram-negative (*E.*
*coli* and *Acinetobacter baumannii* (*A.*
*bau*)) MIC protocol and tuberculosis luminescence assay. Among these compounds, the methyl substitution of R_6_ and ortho substitution of R_5_ on the secondary benzylic amine significantly increased the antimicrobial activities (Figure 18). In particular, the para- or meta-methyl substitution of R_8_ provided the most potent antimicrobial compounds, **66** and **67,** with MIC values of 33 μM (*E.*
*coli* WT), less than 1 μM (*E.*
*coli* Δ-TolC), and 33 μM (*A.*
*bau*) against Gram-negative bacteria as well as *Mycobacterium tuberculosis* (MTB) activity of 10 μM against H37Rv WT. Furthermore, they showed that combination treatment of compound **62** (R_1_-_6_ = -H, R_7_ = -1-cyclohexene, HCl salt) and AN3365 (an LeuRS inhibitor) dramatically decreased the viability of bacterial cells, suggesting that the combination therapy could attenuate antibiotic resistance, thus expanding the lifespan of antibiotics.

### 3.5. Prolyl-tRNA Synthetase (ProRS) Inhibitors

Daewoong Pharmaceutical Co. unveiled heterocyclic compounds to inhibit abnormal human prolyl-tRNA synthetase (ProRS) activity for the treatment of cancers, inflammatory diseases, autoimmune diseases, and fibrosis [59]. A total of 230 nitrogen-containing heteroaromatic compounds with various substitutions were synthesized and categorized into three groups in terms of their biological activities to inhibit the human ProRS enzyme, as shown in Figure 19. The IC_50_ values of 26 compounds (class A) were 100 nM or less, the values of 89 compounds (class B) ranged from 100 to 500 nM, and the values of 115 compounds (class C) were 500 nM or higher. Compounds in class A shared disubstituted imidazolyl aromatic bicyclic rings, yielding potent human ProRS inhibition.

### 3.6. Multi-tRNA Synthetase Inhibitors

The Qi group at the University of Florida prepared 23 sulfamide adenosine derivatives as aminoacyl-tRNA synthetase inhibitors for the treatment of proliferative diseases such as cancers, inflammatory diseases, and autoimmune diseases [60]. As described in Figure 20, a protected adenosyl amine was extended as the protected adenosyl sulfamide, of which the terminal amine was coupled with the different kinds of amino acids to complete the synthesis of the aminoacyl sulfamide adenosine compounds (aa-SA). The cell viability assay of the aa-SAs was performed in prostate cancer cell lines, and IC_50_ values of 0.0015–27 μM for LNCaP and 0.0011–16 μM for PC3 were observed.

For L-Gln-lactam-SA (**89**), a panel of other cancer cell lines was tested for cell viability to give cytotoxic activities (IC_50_) of 0.19–25.24 μM (Table 2). Additionally, apoptotic staining and cell cycle/arrest assays were performed on LNCaP and PC3 cells for all aa-SA compounds. aa-SA-induced apoptosis and a shift of the cell cycle to the sub G0/G1 and S phases were observed (Table 2).

## 4. Aminoacyl-tRNA Synthetase (ARS) Therapeutics in Clinical Trials

Since FDA approval of the natural product mupirocin (an IleRS inhibitor) to treat bacterial infections, prokaryotic ARS has attracted much attention as one of the most promising targets for developing antimicrobial therapeutics in recent years. Nevertheless, AN2690 (a synthetic small molecule, tavaborole) is the only FDA-approved drug for nail onychomycosis to inhibit fungal LeuRS. In this section, we discuss small molecule therapeutics that are undergoing clinical trials as prokaryotic or eukaryotic aminoacyl-tRNA synthetase (ARS) inhibitors. A recent review published in *Nature Reviews Drug Discovery* (2019) by the Kim group at the Medical Bioconvergence Research Center described the ARS inhibitors that are currently being evaluated in clinical trials [14]. Thus, this review will provide the latest updates on the clinically active compounds based upon their reported clinical outcome. The Cortellis search (2020-10-08) with the keywords “tRNA synthetase inhibitor” and status of “phase 1 clinical”, “phase 2 clinical”, “phase 3 clinical”, and “clinical” revealed that five small molecule therapeutics, including DWN12088, CRS3123, GSK3036656, mupirocin bioadhesive gel, and DWP17011, are being actively evaluated in clinical studies (Table 3). Since mupirocin has already been marketed for bacterial skin infection and information on DWP17011 is fairly limited, clinical trials of the remaining three compounds will be discussed in this section.

### 4.1. DWN12088, Eukaryotic Prolyl-tRNA Synthetase (eProRS) Inhibitor

Excessive deposition of collagen is pathologically considered to be the main cause of fibrosis, and proline residues are the major constituents of collagen. Therefore, an eProRS inhibitor would downregulate collagen synthesis and would be a potential therapeutic agent against fibrosis. Daewoong Pharmaceutical Co. Ltd. developed DWN12088 (**91**) (Figure 21), one of the class A compounds (eProRS IC_50_ = 78 nM) covered in the aforementioned patent (WO 2018147626 A1), for the treatment of idiopathic pulmonary fibrosis (IPF). In IPF patient-derived primary lung fibroblasts, treatment with **91** reduced the synthesis of collagen and α-smooth muscle actin (α-SMA) under transforming growth factor-β (TGF-β) induction. In the mouse model of bleomycin-induced pulmonary fibrosis, oral administration of **91** for 2 weeks significantly reduced cardiac fibrosis (ED_50_ = 0.4 mg/kg, in vitro) and the collagen amount in lung tissues and bronchoalveolar lavage fluid (BALF) [61]. Remarkably, **91** provided considerable repair of lung function in this mouse model, as measured by whole-body plethysmograph. GLP (good laboratory practice) toxicity studies of **91** with a 4-week repeated dose in rats and monkeys were conducted to examine a wide safety margin. Furthermore, **91** was designated an orphan drug for IPF by the FDA in August 2019 [62], and a phase I clinical trial (a randomized, parallel-assignment, double-blind, placebo-controlled; ACTRN12619001239156) began in Australia in September 2019. The goal of the study was to assess the safety, tolerability, and pharmacokinetic properties of single- and multiple-ascending doses of **91** compared with placebo following oral administration in healthy volunteers (expected *n* = 80) [63]. The study was conducted in two parts: part A (single-ascending dose, six cohorts, eight subjects/cohort) and part B (multiple-ascending dose, four cohorts, eight subjects/cohort). In part A, two subjects in each cohort received placebo, and six subjects received a single dose (100 mg, 200 mg, 500 mg, or 800 mg) of **91**. In part B, two subjects in each cohort were given placebo, and six subjects received multiple doses (25 mg, 75 mg, 150 mg, or 300 mg) every 12 h for 14 days. The safety and tolerability of **91** patients were evaluated in each cohort prior to dose escalation. Blood samples for PK and PD (pharmacodynamics) analysis were collected at certain time points from the pre-dose of **91** through 72 h post-dose. The preliminary results showed that human ProRS was significantly overexpressed compared with normal lung tissue from IPF patients [61]. This phase I study was expected to be completed in October 2020, and more detailed clinical results are expected to be reported soon.

### 4.2. CRS3123, Prokaryotic Methionyl-tRNA Synthetase (pMetRS) Inhibitor

Crestone, under license from Cardiovascular Systems, developed the antibiotic CRS3123 (**92**) (Figure 21) (formerly REP3123), a prokaryotic methionyl-tRNA synthetase (pMetRS) inhibitor, for the oral treatment of *Clostridium difficile* infection (CDI) and *Helicobacter pylori* infection (HPI) [64]. Notably, **92** prevents protein synthesis by inhibiting *C**. difficile* MetRS with high specificity (*K_i_* = 0.020 nM) and high selectivity (>1000-fold over human MetRS). The in vitro efficacy of **92** was estimated against numerous clinical isolates of *C**. difficile* and epidemic strains BI/NAP1/027 (*n* = 108), showing MIC values of 0.25–1 mg/L [65]. While **92** did not exhibit activity against most Gram-negative bacteria, it had high potency (MIC_90_ < 1 mg/L) against clinically important Gram-positive bacteria, including *S. aureus*, *Streptococcus pyogenes*, *E. faecalis*, and *E. faecium*, which are resistant to current antimicrobial agents. In a clindamycin-induced hamster model of CDI, **92** demonstrated superior efficacy to vancomycin, reducing the production of toxins and spores from *C. difficile* [66,67]. Consequently, **92** was suggested as a promising candidate for the treatment of CDI with special features, including (i) a novel mode of action with maintaining activity against pre-existing resistances; (ii) high selectivity and potency against a narrow spectrum of *C. difficile*, while not affecting normal intestinal microbiota; and (3) the inhibition of toxin production and spore formation from *C. difficile*, alleviating the severity and spread of the disease [68].

#### 4.2.1. Phase I Trial of a Single Dose of CRS3123 (NCT01551004)

In May 2012, the first-time-in-human (FTIH), randomized, double-blind, placebo-controlled, dose-escalation phase I clinical study (NCT01551004) was initiated in the US to assess the safety and pharmacokinetics of **92** after a single oral dose in healthy adults. Five cohorts of eight subjects received **92** or placebo each in a 3:1 ratio [69]; **92** was supplied in 100 and 200 mg capsules, and cohorts A through E received a single oral dose of 100 mg, 200 mg, 400 mg, 800 mg, and 1200 mg, respectively. The clinical data reported by Crestone Inc. in 2017 showed that no serious adverse events or immediate allergic reactions were observed during the study [70]. In the **92**-treated groups, hemoglobin reduction, headache, and abnormal urine analysis were the most frequent adverse events, which were mild to moderate and dose-independent. The increase in the absorption rate of **92** was less than dose proportional, and the relative bioavailability of the 1200 mg dose group was lower than that of the 800 mg dose group. The phase I study showed the safety and tolerability of **92** within the dose ranges and led to further investigation of CRS3123 for the treatment of CDI.

#### 4.2.2. Phase I Trial to Determine the Safety and Pharmacokinetics of CRS3123 (NCT02106338)

An additional single center, randomized, placebo-controlled, double-blind, multiple-ascending dose phase I study (NCT02106338) was pursued to evaluate the safety and tolerability of escalating doses of **92** by oral administration to healthy adults [71]. In this study, thirty healthy subjects aged 18–45 years were randomized into three cohorts, receiving either oral doses of 200, 400, and 600 mg **92** (eight subjects per cohort) or placebo (two subjects per cohort) every 12 h for 10 days. The safety and tolerability and the plasma, urine, and fecal concentrations and systemic exposure of **92** were determined after multiple oral doses for 10 days. The total duration of this study was 46 weeks. Recently, Crestone Inc. reported the results of this clinical study, showing that **92** was generally safe and well-tolerated without serious adverse events (SAEs), severe treatment-emergent adverse events (TEAEs), and any clinically significant cardiac symptoms [68]. Pharmacokinetically, **92** was absorbed rapidly (T_max_ = 1–2 h), and the half-life in plasma was approximately 5–6 h without accumulation after multiple dosing. The increase in plasma concentrations was less than proportional to the increase in dose. Importantly, the high oral dose of **92** was not fully absorbed, and fecal concentrations of all dosages were considerably above the target MIC_90_ value of 1 mg/L for pharmacodynamic success. Moreover, **92** was inactive against important commensal anaerobes, such as *Bacteroides*, *bifidobacteria*, and *clostridia*, showing favorable microbiome data over other current medications for CDI. The results of the two phase I clinical studies described above assured further clinical evaluation of **92** patients with CDI. According to the Cortellis report (Clarivate Analytics, 2020), a phase II clinical study to evaluate **92** in CDI is being planned, and preclinical evaluation for gastric *H. pylori* infection (HPI) with or without a proton pump inhibitor is underway [64].

### 4.3. GSK3036656, Prokaryotic Leucyl-tRNA Synthetase (pLeuRS) Inhibitor

Prokaryotic leucyl-tRNA synthetase (pLeuRS) plays a crucial role in bacterial protein synthesis and has become one of the major targets for the development of antimicrobial therapeutics. In particular, boron-containing compounds have been investigated as LeuRS inhibitors by GlaxoSmithKline. GSK has been developing GSK3036656 (**8**) (Figure 4) (other names, GSK656 and GSK070) as a pLeuRS inhibitor for the treatment of *M. tuberculosis* infection (MTBI), in-licensed from Anacor Pharmaceuticals. Notably, **8** inhibited *Mtb* LeuRS (IC_50_ = 0.20 μM), showing in vitro antitubercular activity (*Mtb* H37Rv, MIC = 0.02 μM) with high selectivity over human mitochondrial and cytoplasmic LeuRS (IC_50_ of > 300 μM and 132 μM, respectively) [21]. Moreover, **8** exhibited remarkable PK profiles in mice (AUC^PO^ [area under the curve of oral administration] = 2.94 h·μg/mL with ~100% BA [bioavailability], Cl [clearance] = 8.5 mL/min/kg, *t*_1/2_ = 3.6 h) and in vivo efficacy (ED_99_ = 0.4 mg/kg) for MTB treatment in acute and chronic TB infection mouse models with superior tolerability (well-tolerated at 300 mg/kg). The preclinical data of **8** showed that it had a tolerable therapeutic window based on a number of preclinical safety and risk assessment studies, as disclosed at the 26th ECCMID in Amsterdam, the Netherlands, in April 2016 [72]. The human efficacious dose was predicted in the range of 0.5–2.7 mg/kg/day, which is lower than that of other antitubercular agents.

#### 4.3.1. First-Time-in-Human (FTIH) Safety and Pharmacokinetics (PK) Study of GSK3036656 in Healthy Subjects (NCT03075410)

In March 2017, a randomized, single-group interventional, double-blind, placebo-controlled, phase I, first-time-in-human (FTIH) study (NCT03075410) to evaluate the safety, tolerability, and pharmacokinetics of single and repeated doses of **8** in healthy participants (expected *n* = 58) was launched in the UK [73]. The study was conducted in two parts, including part A (single dose, two cohorts, nine subjects/cohort) and part B (repeat dose, four cohorts, ten subjects/cohort). In part A, single doses (5 mg, 15 mg, 25 mg, or placebo) of **8** were administered orally. In part B, repeat doses (5 mg, 15 mg, or placebo) of **8** were administered orally once a day for 14 days. The increase in the absorption rate was proportional to the increase in both the single dose and 14-day repeat doses [22]. Based on the PK parameters C_max_ and AUC, accumulation with repeated administration increased approximately 2- to 3-fold, and food intake did not influence the PK parameters. Notably, **8** was not easily metabolized in plasma, but approximately 90% of **8** and 10% of deboronated metabolites by oxidation were detected in urine. Based on drug-related materials detected in urine, the absorbed amounts of **8** from single (25 mg) and repeat (15 mg) dosing were at least 50% and 78%, respectively. From this phase I clinical study completed in August 2017, the safety and tolerability of **8** was acquired after single and multiple doses with no reports of serious adverse events. Importantly, the results from the clinical laboratory, vital signs, electrocardiogram (ECG), and telemetry presented no cardiotoxicity concerns. Clinical trial simulations (CTS) were performed to predict the dose range of **8** that produces the highest possible early bactericidal activity (EBA_0–14_) for the phase II clinical trial, and 10 to 15 mg was suggested to be the optimal dosage of **8** for TB treatment.

#### 4.3.2. An Early Bactericidal Activity, Safety, and Tolerability Study of GSK3036656 in Subjects with Drug-sensitive Pulmonary Tuberculosis (NCT03557281)

In March 2019, an open-label, randomized, sequentially assigned, dose-escalation, phase IIa study (NCT03557281) was initiated to evaluate the bactericidal activity, safety, and tolerability of **8** compared to RIFAFOUR e-275 (Sanofi-Aventis) in patients (expected *n* = 80) with drug-sensitive pulmonary tuberculosis in South Africa [74]. This study was designed to investigate the early bactericidal activity (EBA), safety, and tolerability of **8** in four cohorts of subjects with rifampicin-susceptible tuberculosis. The primary objective of this dose-escalation study was to estimate the antituberculosis effect of **8** by counting serial colony-forming units (CFUs) of MTB in sputum over 14 days. Subjects in each cohort will be treated with either **8** or a standard-of-care (RIFAFOUR e-275) regimen in a 3:1 ratio. The duration of this study for an individual subject was planned to be 5 weeks: 1 week of screening, 2 weeks of treatment, and another 2 weeks of follow-up visit. The reporting of proof-of-concept data was delayed to the first half of 2021 due to the current COVID-19 outbreak [72].

## 5. Conclusions

Nie et al. summarized the roles of ARSs in various immune diseases, such as autoimmune diseases, infectious diseases, and tumor immunity, and described their role as important regulators and signaling molecules in the development of immune cells [75]. Antisynthetase syndrome (ASSD) is a heterogeneous group of autoimmune diseases, including interstitial lung disease (ILD), myositis, mechanic’s hands, Raynaud’s phenomenon, and arthritis. ASSD is often caused by specific anti-ARS autoantibodies [76], and the specificity of anti-ARS autoantibodies is related to the clinical features, disease severity, and survival of ASSD patients [77,78]. ARSs are also dysregulated in other autoimmune diseases, including multiple sclerosis, rheumatoid arthritis, immune thrombocytopenia, and systemic lupus erythematosus [79,80,81,82]. For example, Wang et al. analyzed the interaction of indoleamine-2,3-dioxygenase (IDO)-expressing dendritic cells (DCs) and TrpRS-expressing CD4(+) T cells in Graves’ disease (GD) patients. In general, IDO and tryptophanyl-tRNA synthetase (TrpRS) are responsible for the metabolism and utilization of tryptophan, respectively, and play important roles in immune regulation [83]. From their analysis, the ratio of serum kynurenine to tryptophan was increased in GD, TrpRS expression from CD4(+) T cells was increased, and there was resistance to IDO-mediated immunosuppression from DCs [84]. ARSs are also closely related to tumor immunity, that is, tumor cells can regulate the functions of immune cells by secreting ARSs, or tumor-related immune cells can secrete ARSs and affect tumor development. For example, Wellman et al. suggested that the direct correlation of ThrRS levels with disease stage in ovarian cancer patients is derived from ThrRS-regulated angiogenesis and immune cell responses [85]. Additionally, Adam et al. discovered that cancer cells could upregulate TrpRS in two different ways to adapt to the nutritional stress caused by tryptophan degradation. First, tryptophan depletion caused by the high expression of indoleamine-2,3-dioxygenase-1 (IDO1) and tryptophan-2,3-dioxygenase (TDO2) in LN229 glioblastoma cells upregulates TrpRS expression through the activation of the general control nonderepressible-2 (GCN2) kinase and phosphorylation of eukaryotic translation initiation factor 2α (eIF2α), followed by the activation of activating transcription factor 4 (ATF4). Second, tumor-infiltrating T cells jointly induce the expression of IDO1 and TrpRS in breast cancer, colon carcinoma, and B-cell lymphoma by secreting IFN-γ [86]. As far as we understand, no small molecule directly related to the autoimmune diseases caused by ARSs is known. Given the high limitations of current immuno-oncology therapies in the clinic, ARS-regulating small molecules that are related to tumor immunity can also be an interesting area to explore. The important functions of tRNA synthetases in neurological and neuromuscular disorders have been explored recently as well [87]. ARSs have been studied as therapeutic targets predominantly for pathogen-derived infectious diseases because of concerns that ARS catalytic site inhibitors would block translation in normal cells. However, their indispensable role in other human diseases has been explored, and some ARS inhibitors are undergoing clinical studies for fibrosis as an example. Given the vast number of studies reported on the role of ARS in tumors, it is reasonable to expect that more clinical studies will emerge for cancer therapeutics in the near future. The role of ARS in central nervous system disorders has recently been revealed. As such, the relatively unexplored area of ARS as a target for human diseases other than infectious diseases is being explored by burgeoning biological investigations. Upon these advancements in the newly found roles of ARSs in human diseases, approaches using small molecules targeting ARSs may drive innovative next-generation therapeutic agents.

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
