# Peer review of "Recent Development of Aminoacyl-tRNA Synthetase Inhibitors for Human Diseases: A Future Perspective"

_biomolecules, 2020, doi:10.3390/biom10121625_

Round 1
Reviewer 1 Report
The manuscript by Kim et al. reviews the developments made regarding inhibitors of aminoacyl-tRNA synthetases (aaRSs) when these enzymes, key actors of the translation process, are implicated into human diseases. The review is very comprehensive by showing a wide list of components, retrieved either from the literature or from existing patents. It will be thus of great interest and importance for the scientific community.
However, before this reviewing effort can be used for future perspectives, some key aspects should be discussed and/or clarified. The review mentions at least three cellular processes that the authors wish to see inhibited for therapeutic purposes. The differences between these cellular processes should be better considered and explained. First, the antimicrobial strategy aims at specifically inhibiting the aaRSs from pathogens and targeting species-specific structural or functional peculiarities of the aaRS, in order not to inhibit the aaRS from the host. The purpose here is to eradicate the pathogen.Second, the authors consider the situations where human aaRSs are involved into cancer developments. They list a series of compounds, which have been shown to have cellular anti-tumoral effects. However, in most of these cases, no direct impact of these compounds on the aaRS itself could be demonstrated. Can the authors thus mention these compounds as aaRSs' inhibitors? Moreover, assuming that the inhibitors indeed directly target the aaRS, does such a strategy really possible knowing that the aaRSs are essential and ubiquitous enzymes? Unless the involvement of the aaRS in cancer development is the result of structural or a functional changes of the aaRS that could be directly targeted by an inhibitor? But these aspects are not discussed.
Lastly, regarding aaRS-related central nervous system disorders. These disorders are due to monogenic mutations. It is thus unclear why they are mentioned in a review focused on inhibitors. Does some studies have discovered inhibitors that would specifically hinder the mutated version of the aaRS and then cure the disease?
All the above-mentioned aspects should be more explicitly discussed in the text to avoid any mis-interpretation. Maybe the authors wish to consider the inhibition of a cellular process in which an aaRS is involved, and not to inhibit the aaRS itself (if this is the case, the title of the review should be modified in agreement). Incongruence is also noticeable in the abstract, which appears as a disjointed list of implications of aaRS into distinct human diseases but does not bring out the logic, which will allow the authors to propose future perspectives. Likewise, the conclusion appears disconnected from the content of the article since it brings an additional element of discussion that concerns the implication of aaRSs into autoimmune diseases. It is therefore essential that some rewriting work will be done to clarify these points.
Minor comments:
- The disparity in the writing is visible in the repetitions. For instance, the aaRS abbreviation is defined many times in the text
- In the abstract, the sentence "aaRS are essential enzymes for translating amino acids for protein synthesis" should be reconsidered; amino acids are not translated.
Author Response
Dear Reviewer,
Thank you for your comments and suggestions.
[Reviewer 1]
Referee 1 gave three large comments that require our responses. The authors hope that our response give below can satisfy the comments of reviewer 1.
First, the antimicrobial strategy aims at specifically inhibiting the aaRSs from pathogens and targeting species-specific structural or functional peculiarities of the aaRS, in order not to inhibit the aaRS from the host. The purpose here is to eradicate the pathogen.
- In the introduction, it is mentioned that “ARSs have long been studied as therapeutic targets for pathogen-derived infectious diseases predominantly because of concerns that ARS catalytic site inhibitors would block translation in normal cells.” We have discussed aaRS inhibitors for infectious diseases under this notion. The authors hope that reviewer 1 is satisfied with this phrase for clarifying the goal of developing antibiotics targeting aaRSs.
Second, the authors consider the situations where human aaRSs are involved into cancer developments. They list a series of compounds, which have been shown to have cellular anti-tumoral effects. However, in most of these cases, no direct impact of these compounds on the aaRS itself could be demonstrated. Can the authors thus mention these compounds as aaRSs' inhibitors? Moreover, assuming that the inhibitors indeed directly target the aaRS, does such a strategy really possible knowing that the aaRSs are essential and ubiquitous enzymes? Unless the involvement of the aaRS in cancer development is the result of structural or a functional changes of the aaRS that could be directly targeted by an inhibitor? But these aspects are not discussed.
- In section 2.4. Benzothiazoles, it is discussed that YH16899 does not bind to the active site of pathogen-derived LysRS, but specifically binds to human LysRS without affecting the catalytic activity of LysRS. It is also suggested that LysRS binds to 67LR in the plasma membrane and enhances cell migration that causes cancer metastasis. Based on this finding, it is demonstrated that directly inhibiting the interaction between LysRS and 67LR by YH16899 could suppress cancer metastasis and derive a new class of antimetastatic cancer therapy.
Lastly, regarding aaRS-related central nervous system disorders. These disorders are due to monogenic mutations. It is thus unclear why they are mentioned in a review focused on inhibitors. Does some studies have discovered inhibitors that would specifically hinder the mutated version of the aaRS and then cure the disease?
- As mentioned in the section of conclusion, no small molecule therapeutics are found for aaRS-related autoimmune diseases or central nervous system disorders. However, given the vast number of biological investigations regarding the important roles of aaRSs in those less explored disease areas, it seems apparent that target specific aaRS-regulating small molecules for those disease area is the field to exploit to find innovative next-generation therapeutic agents as a medicinal chemist.
Minor comments:
The disparity in the writing is visible in the repetitions. For instance, the aaRS abbreviation is defined many times in the text. --> we could not understand what reviewer 1 intended to point out from this comment. Please elaborate your comment further? Thanks,
In the abstract, the sentence "aaRS are essential enzymes for translating amino acids for protein synthesis" should be reconsidered; amino acids are not translated. --> Corrected as “Aminoacyl-tRNA synthetases (ARSs) are essential enzymes that ligate amino acids to tRNAs and translate the genetic code during protein synthesis.”
Reviewer 2 Report
This review by Kim et al addresses the topic of small molecule inhibitors of aminoacyl-tRNA synthetases (ARSs), a subject of several recent reviews. Even though the promise of ARSs as therapeutic targets has long been appreciated, the number of therapeutics that have actually gained approval as clinically effective treatments is small. However, there has been a burst of recent interest, and these authors have done a reasonably good job in capturing some of the recent promising candidates. In this manuscript, the authors focus on the medicinal chemistry of a number of the lead compounds which target various ARSs, mining data from both reports and patent applications. This is useful for bringing results to the attention of readers that they might not be aware of from only a survey of the published literature. Additionally, the authors have also screened on-line resources to identify compounds that are in active clinical trials, which is also valuable. Thus, while this review is necessarily of a fairly focused nature, it does provide a valuable service for the field.
I have a few comments about specific points:
- The program investigating the TrpRS fragment as an anti-angiogenic lead was dropped by Pfizer a long time ago. Maybe not the best example of an successful biological-based ARS program.
- In their characterization of patents, the authors might want to make clear which of these patents are provisional applications versus actual granted patents by USPTO.
- With regard to the Glycine Sulfamoyl adenosine inhibitor, I find the claim that this compound has no effect on aminoacylation to be highly implausible, owing to the fact that, as indicated by the crystal structure, it clearly binds in the active site pocket where the glycyl adenylate would normally be formed. I think the authors should examine carefully just how carefully this claim is supported in the application. At the very least, they should mention that all other similar sulfamoyl compounds are quite potent inhibitors.
- There are several important recent publications in this area that the authors have overlooked or which were published after this manuscript was submitted. In the interest of completion, these should be added. These include the following: Kato, N., Comer, E., et al. "Diversity-oriented synthesis yields novel multistage antimalarial inhibitors" Nature 2016; Vinayak, S., Jumani, R., " Bicyclic azetidines kill the diarrheal pathogen Cryptosporidium in mice by inhibiting parasite phenylalanyl-tRNA synthetase" Transl. Med. 12, (2020) 30 September 2020.
Author Response
[Reviewer 2]
Referee 2 gave four comments mainly on the patent part.
- The program investigating the TrpRS fragment as an anti-angiogenic lead was dropped by Pfizer a long time ago. Maybe not the best example of a successful biological-based ARS program.
--> Thank you for your valuable comment. The corresponding phrase of “… and a peptide fragment derived from tryptophanyl tRNA synthetase (TrpRS) is of interest as a possible antiangiogenic agent.” is corrected to “… and a peptide fragment derived from tryptophanyl tRNA synthetase (TrpRS) was studied as a possible antiangiogenic agent. However, it is noticed lately that Pfizer has dropped its development of the TrpRS fragment as an anti-angiogenic lead a while ago.” from line 38 –
- In their characterization of patents, the authors might want to make clear which of these patents are provisional applications versus actual granted patents by USPTO.
--> History of each patent was checked and the granted patents were updated with publication number at the end of each reference.
- With regard to the Glycine Sulfamoyl adenosine inhibitor, I find the claim that this compound has no effect on aminoacylation to be highly implausible, owing to the fact that, as indicated by the crystal structure, it clearly binds in the active site pocket where the glycyl adenylate would normally be formed. I think the authors should examine carefully just how carefully this claim is supported in the application. At the very least, they should mention that all other similar sulfamoyl compounds are quite potent inhibitors.
--> The sentence of “Interestingly, GlySAs do not significantly influence the function of GlyRS in aminoacylation.” was deleted for clarity.
4.There are several important recent publications in this area that the authors have overlooked or which were published after this manuscript was submitted. In the interest of completion, these should be added. These include the following: Kato, N., Comer, E., et al. "Diversity-oriented synthesis yields novel multistage antimalarial inhibitors" Nature 2016; Vinayak, S., Jumani, R., " Bicyclic azetidines kill the diarrheal pathogen Cryptosporidium in mice by inhibiting parasite phenylalanyl-tRNA synthetase" Transl. Med. 12, (2020) 30 September 2020.
-->It is agreed that the bicyclic azetidines should be mentioned in the manuscript for completion. Section 2.7 is added for the bicyclic azetidines. The authors thank for this supportive comments of reviewer 2.
Thanks,
Reviewer 3 Report
Aminoacyl-tRNA synthetases (ARSs) are the enzymes that catalyze the aminoacylation reaction by covalently linking an amino acid to its cognate tRNA in the first step of protein synthesis. Beyond this classical function, ARSs are also known to have a role in several metabolic and signaling pathways that are important for cell viability, also called moonlighting functions. It is becoming increasingly clear that the study of these enzymes is of great interest to discover potential drug target against various diseases.
In a first part of their review, the authors listed the structural class of ARS inhibitors. For instance, benzoxaboroles are antimicrobial compounds targeting LeuRS. Cladosporin has antimalarial activity by targeting the parasite LysRS. Similarly, borrelidin targets ThrRS for malaria. LysRS is targeted by benzothiazoles because of its interaction laminin receptor (67LR), a target for antimetastatic therapeutic. Halofuginone, a well-known antifibrotic agent in preclinical and clinical studies, inhibits prolyl-tRNA synthetase (ProRS) activity. The AIMP2 scaffold protein of the multi-synthetase complex is dissociated from the complex by sulfonamido propanamides influencing the activity of the p53, TGF-β, TNF-α and WNT signaling pathways.
In the following section, the authors discussed patents related to therapeutic development targeting both prokaryotic and eukaryotic ARSs. Human ARS are becoming major targets in treatment of proliferative diseases such as cancers, inflammatory diseases, autoimmune diseases and fibrosis. These include GlyRS inhibitors for cancer therapy, LeuRS and MetRS inhibitors for antibacterial treatment, PheRS inhibitors with antimicrobial activity, ProRS inhibitors for the treatment of cancers, inflammatory diseases, autoimmune diseases, and fibrosis, and multi-tRNA synthetase inhibitors derived from sulfamide adenosine compounds.
In the last section, the authors summarize the clinical trials based on small molecule inhibitors that target prokaryotic or eukaryotic aminoacyl-tRNA synthetase. As a recent review was already published by the Kim group on the same topic in 2019, the authors provided the latest updates on the clinically active compounds based upon their reported clinical outcome. These include 5 compounds DWN12088, CRS3123, GSK3036656, mupirocin bioadhesive gel, and DWP17011 that target ProRS, MetRS, LeuRS, IleRS.
Altogether the review offers comprehensive information on the available inhibitors against ARSs. Recent developments of ARS inhibitors are discussed, starting from their discovery and continuing with clinical trials. The literature is accurately cited (articles, clinical trials identifiers) including the reviews focusing ARS-induced disorders (infectious and inflammatory diseases, cancer, CNS disorders). This review is a good inventory for researchers working in this area.
Author Response
Referee 3 mentioned that this review is a good inventory for researchers working in this area, and recommended for publication as it is. Thank you,
Reviewer 4 Report
Biomolecules-1001146
Minsoo Song et al. provided an overview on the recent development of aminoacyl tRNA synthetase (aaRS) inhibitors for human diseases. The paper specifically looks at more recent developments and specifically refers to previous published reviews for older findings.
In the past a lot of work on aaRS inhibitor development was mainly focused on controlling pathogen-derived infectious diseases, which has led to development of small molecule therapeutics. However, especially in the industrial setting, increasing attention is given to the applicability of ARS inhibitors for other human diseases such as fibrosis, cancers or CNS disorders, in view of the many different roles that are bestowed on specific human aaRS in control of various biochemical events. The present review takes also these applications into account and overall gives a nice overview of the recent developments in the aaRS field as intended. Therefore, in essence the manuscript is worthwhile publishing, provided a revision considers the relatively large number of shortcomings that should be remedied, as detailed below.
Detailed comments:
Overall, at several places the language is sloppy and being imprecise – see below.
In addition, several figures are of very poor quality and not acceptable for publication (e.g. figure 4 – especially all text parts; but many others as well like fig.6, 7, 8 and so on). In some cases the chemical drawings themselves are of poor quality, but text parts in almost all figures need quality improvement.
Line 50-51 is an almost complete duplication of previous sentence and can be removed.
Line 80: correct to the underlined phrasing “The aminomethyl side chain at C3 in 4 and 5 was critical for binding, and C4-halogen atoms… “
Lines 109-111: please include some 2D or 3D picture to highlight the various interactions mentioned in the text. This makes it easier for the reader to understand the importance of the various parts of the interacting ligand.
Line 122: can the authors comment on the claim that both phenolic hydroxyl groups are critical (as mentioned in line 122), while another study simply leaves out these groups resulting in compound number 21. It is rather awkward to find both statements close to each other without any comment. In general, in a good review authors should regularly express their opinion or thoughts on the reported findings.
Line 131 (and other places): could the authors also indicate in their manuscript which assay was used for IC50 determination of cladosporin analogues (compounds shown in figures 5 and 6).
Line 167: typo to be corrected: “intime” to “in time”
Line 254-255: “4-glycylsulfamoyladenosine” - what does the "4" stand for?
line 256-258: it is unclear what the authors mean in this part; please rephrase this part of the sentence. (covalent attachment of neural cells?)
Line 263: binding of GlySA to the active site of GlyRS is shown by the respective crystal structure (see ref 31). While all other aaSAs are known as strong inhibitors of aminoacylation, this would not be the case for GlySA? Has this been documented properly? The authors should either include this information or otherwise give their opinion in this review on this deviating compound.
Line 267: explain “GI50 values”
Line 290: in this latter part of the review the authors are focusing on what they found in patent literature. However, the specific results as referred to in ref 49, can be found in literature as well, and should be quoted (ACS Med. Chem. Lett. 2018, 9, 84−88). The potential availability of analogous information to be found in scientific journals should be checked for the other cited patents as well.
Line 296-297: please revise the sentence “However, none of the compounds presented any proper antibacterial activities at concentrations of 50 uM or below against IleRS and ValRS from E. coli and S. aureus.” I presume the authors mean either “antibacterial activities against E coli” or “enzyme-inhibitory activities against IleRS and ValRS”.
Line 298: please correct this awkward sentence: “It was found that the valine of the amino acid … increased the inhibition of LeuRS” (?)
Line 300: compound “42” should be corrected to “compound 40” according to figure 13 - please check the original reference. Are the mentioned values EC50 values for bacterial growth inhibition or IC50 values for LeuRS inhibition? Overall, the authors should be much more precise in their language.
Line 326: add compound numbers to the description: “…carboline compounds (45-47)…”.
Line 353: activities for compound 65 and its structure should be added to figure 16.
Line 368: have any antitumoral data or other biological activity data been unveiled? If so, this information should be added.
Line 443: throughout the whole paragraph (line 443-458) compound number 90 each time should be noted in bold.
Line 469-472: the logic is missing in this statement: with increasing dose the metabolism (glucuronidation) might be lagging if we encounter a saturated process; in such case higher doses should have a slightly higher relative bioavailability instead (as of reduced relative metabolism), in contrast to the actual statement. If this interpretation is wrong, the authors should revise their arguments.
Author Response
[Reviewer 4]
Referee 4 made nine critical comments, which we address as described below.
- In addition, several figures are of very poor quality and not acceptable for publication (e.g. figure 4 – especially all text parts; but many others as well like fig.6, 7, 8 and so on). In some cases the chemical drawings themselves are of poor quality, but text parts in almost all figures need quality improvement.
- --> All figures and texts are rearranged in the same scale.
- Line 50-51 is an almost complete duplication of previous sentence and can be removed.
- --> Line 50-41 is deleted.
- Line 80: correct to the underlined phrasing “The aminomethyl side chain at C3 in 4 and 5 was critical for binding, and C4-halogen atoms… “ --> corrected.
- Lines 109-111: please include some 2D or 3D picture to highlight the various interactions mentioned in the text. This makes it easier for the reader to understand the importance of the various parts of the interacting ligand. --> structural aspect is discussed at the end of section 2.2 Cladosporin with Figure 7 (Binding modes of cladosporin and 20 to Pf LysRS1).
- Line 122: can the authors comment on the claim that both phenolic hydroxyl groups are critical (as mentioned in line 122), while another study simply leaves out these groups resulting in compound number 21. It is rather awkward to find both statements close to each other without any comment. In general, in a good review authors should regularly express their opinion or thoughts on the reported findings.
- #4~5: Figure 7 is added show the binding mode of cladosporin and 20 with Pf An analysis for their binding modes is given from line 151 to 165 as well.
- Line 131 (and other places): could the authors also indicate in their manuscript which assay was used for IC50 determination of cladosporin analogues (compounds shown in figures 5 and 6).
- For compounds 12-14, Transcreener AMP assay system to assess aminoacylation by LysRS is indicated with its reference in line 112. For compounds 15-19, ATP hydrolysis assay is used and it is already indicated in the context. For compounds 20-21, luciferase ATP consumption assay is used, and is indicated as in “… (IC50 = 0.015 μM, luciferase ATP consumption assay) and …”
- Line 167: typo to be corrected: “intime” to “in time” --> corrected.
- Line 254-255: “4-glycylsulfamoyladenosine” - what does the "4" stand for? --> corrected as “the four different glycylfulfamoyladenosine”
- line 256-258: it is unclear what the authors mean in this part; please rephrase this part of the sentence. (covalent attachment of neural cells?) --> corrected as “The covalent attachment of NEDD8 (neuronal precursor cell-expressed developmentally down-regulated gene 8)”
- Line 263: binding of GlySA to the active site of GlyRS is shown by the respective crystal structure (see ref 31). While all other aaSAs are known as strong inhibitors of aminoacylation, this would not be the case for GlySA? Has this been documented properly? The authors should either include this information or otherwise give their opinion in this review on this deviating compound.
--> The sentence of “Interestingly, GlySAs do not significantly influence the function of GlyRS in aminoacylation.” was deleted for clarity.
- Line 267: explain “GI50 values”
--> sentence of “GlySA (31) possessed a potency of 0.372 to 3.715 mM GI50 values in 58 cancer cell lines.” was corrected as “GlySA (31) showed a range of growth inhibition values (GI50 0.372~3.715 mM) in 58 cancer cell lines.”
- Line 290: in this latter part of the review the authors are focusing on what they found in patent literature. However, the specific results as referred to in ref 49, can be found in literature as well, and should be quoted (ACS Med. Chem. Lett. 2018, 9, 84−88). The potential availability of analogous information to be found in scientific journals should be checked for the other cited patents as well.
--> The journal was added as a new reference and all the other reference numbers were corrected accordingly. Also, the other patents were checked for information on scientific journals and two other journals were added correctly.
- Line 296-297: please revise the sentence “However, none of the compounds presented any proper antibacterial activities at concentrations of 50 uM or below against IleRS and ValRS from E. coli and S. aureus.” I presume the authors mean either “antibacterial activities against E coli” or “enzyme-inhibitory activities against IleRS and ValRS”.
--> corrected as “However, none of the compounds presented any inhibitory activities at concentrations of 50 mM or below against IleRS and ValRS bacterial enzymes from either E. coli or S. aureus.
- Line 298: please correct this awkward sentence: “It was found that the valine of the amino acid … increased the inhibition of LeuRS” (?)
--> corrected as “It was found that the compounds bearing meta-substituted phenyl ring as well as valine moiety presented the significant increase in the inhibition of LeuRS”
- Line 300: compound “42” should be corrected to “compound 40” according to figure 13 - please check the original reference. Are the mentioned values EC50 values for bacterial growth inhibition or IC50 values for LeuRS inhibition? Overall, the authors should be much more precise in their language.
--> corrected as “IC50 values of compound 40: 0.002 mM for E. coli LeuRS and 0.33 mM for S. aureus LeuRS”
- Line 326: add compound numbers to the description: “…carboline compounds (45-47)…”.
--> numbers are added
- Line 353: activities for compound 65 and its structure should be added to figure 16.
--> compound 65 was corrected as compound 62 in Figure 18 and all the compound numbers are re-numbered accordingly.
- Line 368: have any antitumoral data or other biological activity data been unveiled? If so, this information should be added.
--> There is no any antitumoral data or other biological activity data given in the reference.
- Line 443: throughout the whole paragraph (line 443-458) compound number 90 each time should be noted in bold.
--> corrected as bold.
- Line 469-472: the logic is missing in this statement: with increasing dose the metabolism (glucuronidation) might be lagging if we encounter a saturated process; in such case higher doses should have a slightly higher relative bioavailability instead (as of reduced relative metabolism), in contrast to the actual statement. If this interpretation is wrong, the authors should revise their arguments.
--> "... partially due to the metabolism of glucuronidation at amino groups of 90” in Line 470-471 was deleted.
Thank you,
Round 2
Reviewer 4 Report
The review manuscript of Minsoo Song et al. on the recent development of aminoacyl tRNA synthetase (aaRS) inhibitors for human diseases has been thoroughly updated and corrected taking into account the various remarks of the reviewers. Apart from the lower quality figures, only some minor remarks are formulated here below.
Line 44: “For this review” instead of “From this review”
line 116: the formulated remark regarding addition of a figure was not answered here but further down in the manuscript. Hence add here in brackets “(see also below with figure 7 and its accompanying discussion)”
line 172: correct to “tetrahydropyran ring”
lines 159-179: The added parts comprise a nice color figure (Fig. 7) showing the binding mode of cladosporin and compound 20, and provide adequate explanation, very much clarifying the previous phrasings.
line 280: add space to "thehydroxymethyl"
lines 322-323: the adapted text has become unreadable; please check again.
Line 354: the valine containing compounds do not inhibit LeuRS - this should be rephrased again or left out.
Overall this is a very valuable review.